# DIFFERENTIALLY PRIVATE SGD WITH SPARSE GRADIENTS

## ABSTRACT

To protect sensitive training data, differentially private stochastic gradient descent (DP-SGD) has been adopted in deep learning to provide rigorously defined privacy. However, DP-SGD requires the injection of an amount of noise that scales with the number of gradient dimensions, resulting in large performance drops compared to non-private training. In this work, we propose *random freeze* which randomly freezes a progressively increasing subset of parameters and results in sparse gradient updates while maintaining or increasing accuracy. We theoretically prove the convergence of random freeze and find that random freeze exhibits a signal loss and perturbation moderation trade-off in DP-SGD. Applying random freeze across various DP-SGD frameworks, we maintain accuracy within the same number of iterations while achieving up to 70% representation sparsity, which demonstrates that the trade-off exists in a variety of DP-SGD methods. We further note that random freeze significantly improves accuracy, in particular for large networks. Additionally, axis-aligned sparsity induced by random freeze leads to various advantages for projected DP-SGD or federated learning in terms of computational cost, memory footprint and communication overhead.

## 1 INTRODUCTION

The success of machine learning, and deep neural networks in particular, combined with ubiquitous edge computation and digital record keeping, has led to a surge in privacy sensitive learning applications. Internet-scale data promises to accelerate the development of data-driven statistical approaches, but the need for privacy constrains the amalgamation of such datasets. Private data are in fact isolated, constraining our ability to build models that learn from a large number of instances. On the other hand, the information contained in locally stored data can also be exposed through releasing the model trained on a local dataset (Fredrikson et al., 2015; Shokri et al., 2017), or even reconstructed when gradients generated during training are shared (Zhu et al., 2019; Geiping et al., 2020; Zhu & Blaschko, 2021).

To address these issues, many applications of machine learning are expected to be privacy-preserving. While differential privacy (DP) provides a rigorously defined and measurable privacy guarantee for database operations (Dwork & Roth, 2014), it also contains intriguing properties, such as robustness to post-processing and composability, which enables conveniently computing an overall privacy guarantee for several DP components. Differential privacy[1] defines privacy with respect to the difficulty of distinguishing the outputs. For a pair of neighboring databases $X, X' \in \mathcal{X}$, i.e. $X$ can be obtained from $X'$ by adding or removing an element.

**Definition 1.** *A randomized mechanism $\mathcal{M} : \mathcal{X} \to \mathcal{R}$ is $(\varepsilon, \delta)$-differentially private, if for any subset of outputs $S \subseteq \mathcal{R}$ it holds that:*

$$\Pr[\mathcal{M}(X) \in S] \leq e^{\varepsilon} \Pr[\mathcal{M}(X') \in S] + \delta.$$

A common paradigm for a randomized mechanism $\mathcal{M}$ in deep learning is perturbed gradient descent:

$$\mathcal{M}(X) := f(X) + \mathcal{N}(0, S_f^2 \sigma^2 \boldsymbol{I}), \tag{1}$$

---

[1]In this work we only consider approximate differential privacy which includes the $\delta$ term.

where $f : \mathcal{X} \to \mathcal{R}$ computes an aggregated gradient given a database $X$ or $X'$. The isotropic Gaussian distributed noise $\xi_{DP} \sim \mathcal{N}(0, S_f^2 \sigma^2 \boldsymbol{I})$ is calibrated to $f$'s sensitivity $S_f^2$, which is the maximal $\ell_2$ distance $\|f(X) - f(X')\|$, i.e. the maximal $\ell_2$ norm of gradient among all individual examples. $X, X'$ could be batches of training data, for instance, in our experiments they are batches of image-label pairs. The factor $\sigma$ is a noise multiplier controlling the strength of the privacy guarantee: higher $\sigma$ leads to lower privacy loss. Differentially private stochastic gradient descent (DP-SGD) upper bounds the certainty of connecting data with arbitrary subset of gradient space using the privacy budget variables $(\varepsilon, \delta)$.

Bassily et al. (2014) show that in a convex setting, DP-SGD achieves excess risk of $\tilde{O}(\sqrt{d}/n\varepsilon)$ for a model $w \in \mathbb{R}^d$ that minimizes the empirical risk $\sum_{i=1}^{n} \ell(w, x_i)$, where $x_1, x_2, ..., x_n$ are drawn from $\mathcal{X}$. While we show that in non-convex general setting, the mean square error (MSE) of perturbed gradient $\tilde{g} = g + \xi_{DP}$ is between $\Omega(d)$ and $\Omega(d^2)$ by assuming the gradients follow a Gaussian distribution:

**Theorem 1.** *Assuming that the gradient is drawn from $\mathcal{N}(\nabla w, \Sigma)$, centered at the true gradient $\nabla w$ and with respect to the covariance matrix $\Sigma$ whose trace goes linearly up with dimension $d$. The MSE of perturbed gradient $\tilde{g} = g + \xi_{DP}$ can be lower bounded by:*

$$MSE \geq \text{Tr}[\Sigma](1 + d\sigma^2). \tag{2}$$

from which we conclude that the lower bound on MSE is between linear and quadratic in $d$ in practice. For the proof and conclusion, refer to Appendix A. In terms of deep learning, as $d$ is a large number for modern network architectures, this can lead to a significant increase in error.

The work of Abadi et al. (2016) proposed to clip the gradient of each individual example in $\ell_2$ norm to a preset bound C, i.e. $\bar{g} = g \cdot \min(1, \frac{C}{\|g\|_2})$. They then apply this clipping bound to compute the variance of Gaussian distributed noise. The Gaussian noise mechanism can be expressed as:

$$\mathcal{M}(D) := f(D) + \mathcal{N}(0, C^2 \cdot \sigma^2 \boldsymbol{I}_d). \tag{3}$$

DP-SGD with gradient clipping has been empirically verified to be effective as it constraints the amount of injected noise by setting a small clipping bound. However, clipping removes the magnitude information of the gradient and therefore results in gradient estimation bias. Setting a small clipping bound with a deeper network will not result in better performance. The expected MSE of the perturbed gradient is only constrained to $O(d)$, the impact of perturbation is non-negligible in practice. The biggest network where this strategy is successfully applied so far is a CNN with Tanh proposed by Papernot et al. (2021), which reaches $\sim 66\%$ accuracy on CIFAR10 in a low privacy regime and is regarded as the state-of-the-art (SOTA) end-to-end network with DP-SGD. To address this curse of dimensionality, most recent works concentrate on gradient dimension reduction.

## 1.1 RELATED WORKS

Abadi et al. (2016) propose to pretrain a network on an auxiliary dataset and then transfer the feature extraction, so that only a linear classifier will be replaced and trained on the private data. Tramer & Boneh (2021) adopt ScatterNet (Oyallon et al., 2019) to extract handcrafted features and train a relatively shallow network based on the features. Both work decrease $d$ by excluding the majority of parameters during DP learning, which also constrains the learning ability of network.

Inspired by the empirical observation that the optimization trajectory is contained in lower-dimensional subspace (Vogels et al., 2019; Gooneratne et al., 2020; Li et al., 2020), a line of work intend to reduce $d$ by exploiting the low-rank property of the gradient while considering privacy. Several recent works (Zhou et al., 2021; Yu et al., 2021a; Kairouz et al., 2021) project the gradient into a subspace which is identified by auxiliary data or released historical gradients. In practice, they use the power method to search for the subspace. The computational cost of running the power method and projecting gradients as well as the memory footprint of storing the projection matrix limits the application of such method to large models. Zhang et al. (2021) target an NLP task where networks are heavily over-parameterized and gradients are extremely sparse, and propose to adopt DP selection to privately select top-$k$ significant gradients for optimization. Also targeting an NLP task, Yu et al. (2021b) propose a low-rank reparameterization of weights via released historical gradients.

In addition to the aforementioned works, McMahan et al. (2017); Yang et al. (2019) and others study how to incorporate differential privacy in collaborative training, e.g. federated learning, in the interest of protecting the privacy of participants. Federated learning also suffers from large $d$, as it is usually deployed on edge devices and local models are periodically synchronized, so communication cost becomes expensive both in time and power usage (Pathak et al., 2012). Therefore, gradient dimension reduction can have large benefits in federated learning involving power-restricted edge devices. A line of work studies how to tackle this issue by utilizing the low-rank property (Shokri & Shmatikov, 2015; Yang et al., 2019; Liu et al., 2020).

## 1.2 OUR CONTRIBUTION

In this work, we demonstrate an axis-aligned gradient dimension reduction method. Our work is orthogonal to previous works, we do not extract any characteristic information of the gradient or model, or approximate the gradient from a subspace. Instead, we randomly zero-out a fraction of the gradient during training and force the gradient to have a sparse representation. We provide a theoretical study on this strategy and reveal that random freeze exhibits a trade-off between signal loss and perturbation moderation in DP-SGD. We remark that our theory and approach do not necessarily rely on the low-rank assumption. To the best of our knowledge, we are the first to study this trade-off in DP-SGD and provide an effective approach.

We use the benchmark CIFAR10 (Krizhevsky, 2012) which is to date standard in benchmarking DP learning and show that well implemented random freeze exhibits various advantages and can be widely applied. More specifically, we maintain accuracy when we adapt projected DP-SGD with random freeze, while we reduce the computational cost and memory footprint induced by the power method and projection. Applying it to various frameworks, we achieve a high representation sparsity of gradient without a loss in performance. Federated learning can take advantage of the resulting sparse representation to reduce communication costs. We further note that the random freeze strategy improves the accuracy of large networks, which we demonstrate with the SOTA End-to-end CNN proposed by Papernot et al. (2021).

## 2 ANALYSIS OF RANDOM FREEZE

In this section we theoretically prove that to a certain freeze rate $r$, DP-SGD with random freeze will converge if the approach without random freeze can converge. Then we investigate the trade-off between signal loss and perturbation moderation induced by random freeze. Furthermore, we empirically demonstrate benefits of applying random freeze from the perspective of the gradient distribution.

### 2.1 CONVERGENCE RATE OF DP-SGD WITH RANDOM FREEZE

Let $\mathcal{L}$ be the objective function $\mathcal{L}(w) := \mathbb{E}_{x \sim \mathcal{X}}[\ell(w, x)]$, $m \in \{0, 1\}^d$ the freeze mask and $r$ the freeze rate, we randomly draw $rp$ indices and set these positions in the mask to $0$ and others to $1$ so that $\sum m = (1 - r)d$. We assume an oracle telling us the true gradient $\nabla w$ and individual gradient $g_{t,i} = \nabla w + \xi_t(x_i)$, where $\xi_t$ is independent gradient deviation with zero mean. Let $p_t$ be the distribution of $\xi_t$. For random freeze the sparse gradient is $g'_{t,i} = \nabla w'_t + \xi'_t(x_i)$, where $\nabla w'_t := m \odot \nabla w_t$, $\xi'_t(x_i) := m \odot \xi_t(x_i)$. Denote for DP-SGD the averaged clipped gradient of $B$ samples $\bar{g}_t := \frac{1}{B} \sum_i g_{t,i} \cdot \min(1, \frac{C}{\|g_{t,i}\|})$, and for DP-SGD with random freeze $\hat{g}_t := \frac{1}{B} \sum_i g'_{t,i} \cdot \min(1, \frac{C}{\|g'_{t,i}\|})$. We have for DP-SGD with random freeze the following inequality:

**Theorem 2.** *Assume G-Lipschitz smoothness of $\nabla w$ such that $\|\nabla w_{t+1} - \nabla w_t\| \leq G\|w_{t+1} - w_t\|$. Consider an algorithm with clipping bound $C$, learning rate $\gamma$ and choose a symmetric probability density distribution $\tilde{p}(\cdot)$ satisfying $\tilde{p}_t(\xi_t) = \tilde{p}_t(-\xi_t)$, $\forall \xi_t \in \mathbb{R}^d$. Then $\exists \kappa \geq 1 - r$ such that:*

$$\frac{1}{T} \sum_{t=1}^{T} P_{\xi_t \sim \tilde{p}_t}(\|\xi_t\| < \frac{C}{4}) h(\nabla w_t) \|\nabla w_t\| \leq \frac{1}{\kappa}(\frac{\Delta_{\mathcal{L}}}{\gamma T} + \gamma \Delta_C + (1 - r)\gamma \Delta_{DP} - \frac{1}{T} \sum_{t=1}^{T} \mathbb{E}_m[b_t]), \quad (4)$$

where we define $b_t := \int \langle \nabla w'_t, g'_{t,i} \cdot \min(1, \frac{C}{\|g'_{t,i}\|}) \rangle (p'_t(\xi'_t) - \tilde{p}'_t(\xi'_t)) d\xi'_t, p'_t, \tilde{p}'_t$ are corresponding projected distribution, and define $\Delta_{\mathcal{L}} := \mathbb{E}[\mathcal{L}_1 - \min_w \mathcal{L}(w)]$, $\Delta_C := \frac{GC^2}{2}$, $\Delta_{DP} := \frac{C^2\sigma^2 dG}{2B^2}$, $h(\nabla w_t) := \min(\|\nabla w_t\|, \frac{3C}{4})$. The proof can be found in Appendix B.

When no freeze $r = 0$, i.e. $m$ is a vector filled with value 1. Note that $\kappa = 1$ for $r = 0$, we can obtain the inequality 4 in the following form:

$$\frac{1}{T} \sum_{t=1}^{T} P_{\xi_t \sim \tilde{p}_t}(\|\xi_t\| < \frac{C}{4}) h(\nabla w_t) \|\nabla w_t\| \leq \frac{\Delta_{\mathcal{L}}}{\gamma T} + \gamma(\Delta_C + \Delta_{DP}) - \frac{1}{T} \sum_{t=1}^{T} b_t, \qquad (5)$$

which describes DP-SGD without random freeze. Chen et al. (2020b) argue that by tweaking $\tilde{p}_t$ it is possible to bound $P_{\xi_t \sim \tilde{p}_t}$ away from zero which means the l.h.s. is proportional to $\|\nabla w_t\|$ or $\|\nabla w_t\|^2$ at each iteration, while letting the convergence bias term $-b_t$ tend to be small as $p_t$ is approximately symmetric. They then prove that by setting a certain learning rate $\gamma$, the r.h.s. diminishes to zero, so the network under DP-SGD can converge. Adapted from that, we see at the r.h.s. of inequality 4 for random freeze, $-\mathbb{E}_m[b_t]$ also tends to be small as $p'_t$ is expected to be approximately symmetric if $p_t$ is. Additionally, $1/\kappa$ will not be large as along as $r$ is not extremely close to 1. So with the same learning rate and $\gamma$ adapted for DP-SGD without random freeze, the r.h.s. of inequality 4 will also tend to zero, which proves the convergence of DP-SGD with random freeze.

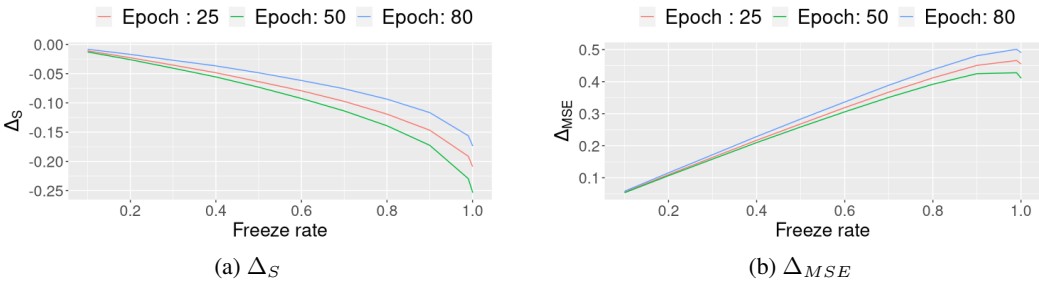

(a) $\Delta_S$        (b) $\Delta_{MSE}$

Figure 1: (a): Signal loss of no freeze minus random freeze. (b): MSE of gradient estimation of no freeze minus random freeze.

## 2.2 TRADE-OFF BETWEEN SIGNAL LOSS AND PERTURBATION MODERATION

From Theorem 2, we see that by applying random freeze the injected noise term $\Delta_{DP}$ is reduced by $\frac{1-r}{\kappa} < 1$ while other terms could be increased by $1/\kappa$. From an operational view, random freeze moderates perturbation while removing some signal. Therefore, the convergence rate of random freeze is a trade-off between signal loss and perturbation moderation. Consider that $\Delta_{DP}$ is proportional to the number of parameters $d$, which is large for neural networks. It might be worthwhile to remove some signal while moderating perturbation. To investigate this question, we consider the MSE of gradient estimation of DP-SGD with and without random freeze at iteration $t$:

$$MSE_{DP} := \mathbb{E}[\|\nabla w_t - (\bar{g} + \xi_{DP})\|^2] = \mathbb{E}[\|\nabla w_t - \bar{g}\|^2] + \frac{C^2\sigma^2 d}{B^2}, \qquad (6)$$

$$MSE_{RF} := \mathbb{E}[\|\nabla w_t - (\hat{g} + m \odot \xi_{DP})\|^2] \overset{32}{=} \mathbb{E}[\|\nabla w_t - \hat{g}\|^2] + (1-r)\frac{C^2\sigma^2 d}{B^2}. \qquad (7)$$

Two statistics of interest are: First, the difference of signal loss $\Delta_S := \mathbb{E}[\|\nabla w_t - \bar{g}\|^2] - \mathbb{E}[\|\nabla w_t - \hat{g}\|^2]$. DP-SGD induces signal noise due to clipping bias while random freeze has additional loss due to freezing. Second, the difference of MSE $\Delta_{MSE} := MSE_{DP} - MSE_{RF}$ which implies the trade-off between signal loss and perturbation moderation. We train the End-to-end CNN with 80 epochs on a privacy budget ($\varepsilon = 7.53, \delta = 10^{-5}$) and measure these two statistics with respect to empirical distribution at the end of three training stages (see Figure 1). The result shows that although random freeze has more signal loss as the freeze rate increasing, due to a large amount of injected noise, MSE of gradient estimation with random freeze is clearly lower. The result reflects

that injected noise is dominant during optimization. Combining this with Theorem 2, it shows the possibility that we achieve better convergence by removing some signal while mitigating noise. For example, in addition to taking advantage of noise mitigation, we can also raise the noise to $\xi_{DP}^+$ so that $\mathbb{E}[\|m \odot \xi_{DP}^+\|]$ matches $\mathbb{E}[\|\xi_{DP}\|]$. This allows us to run more iterations $T$ on the same privacy budget, which also implies better convergence by Theorem 2. In section 4 we will demonstrate that when $d$ is large, which implies large $\Delta_{DP}$, we can improve the performance by applying random freeze.

### 2.3 A GRADIENT DISTRIBUTION VIEW OF RANDOM FREEZE

We also observe the potential benefit of random freeze from the perspective of the gradient distribution. Compared to no freeze, random freeze leads to less clipping bias and gradient distortion, as shown in Figure 2. We adopt the same clipping bound for random freeze and no freeze. As fewer dimensions contribute to the norm computation, random freeze reduces the clipping probability and therefore alleviates clipping bias (Zhang et al., 2020). We also note that the norm of sparse gradients are not equally scaled down, weak gradients can spontaneously become larger during training, which mitigates the distortion of gradients due to perturbation, while the perturbation of random freeze is already moderated compared to no freeze.

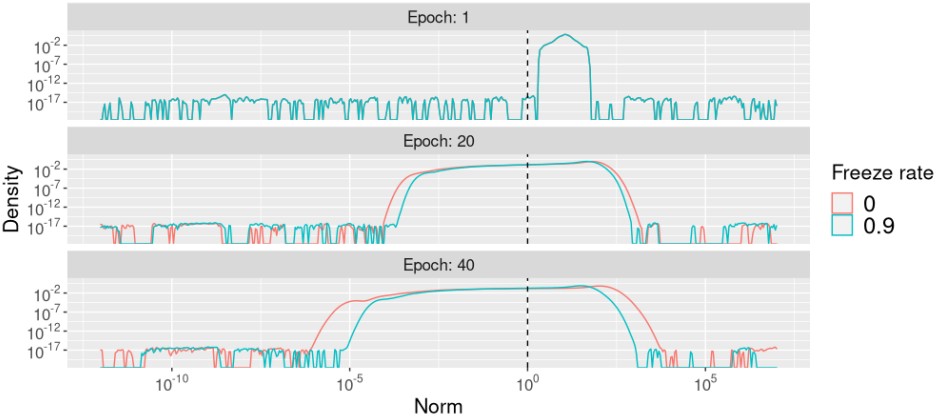

Figure 2: Gradient norm distribution of DP-SGD with or without the random freeze strategy. The vertical dashed line indicates the clipping bound. With the random freeze strategy, in later epochs the variance of the norm magnitude decreases. A lower number of high-magnitude gradient norms implies less clipping bias, while the decrease in low magnitude gradient norms implies a higher signal-to-noise ratio of the perturbed gradients. The two plots overlap in the subfigure corresponding to the first epoch as the freeze rate is 0 and the networks are initialized equally. The freeze rate at the 20th epoch is 0.45 and reaches 0.9 at the 40th epoch. Note that both axes are in log scale.

## 3 TECHNICAL DETAILS

Algorithm 1 outlines our approach. Since the coordinates that have been selected to freeze do not depend on the dataset, their indices can be exposed or transferred in clear text and do not lead to additional privacy loss to the dataset. We apply random freeze to optimization with SGD with or without momentum. In case of non-zero momentum, the velocity is updated with the sparse gradient as normal. That means, parameters that have been frozen can still be updated as long as their velocity has not decayed to zero. Next, we discuss the properties and implementation details of random freeze, compare different variants and provide insight into the strategy,

### 3.1 GRADUAL COOLING

We find that if we initiate the training with constant freeze rate $r$, the network converges slowly and performs poorly when the privacy budget has been fully consumed. The reason is that in the early stages of training, the network is far from its optimal position, it is better to let all parameters stay

---

**Algorithm 1:** Random freeze

---

**Input:** Initialized parameters: $\mathbf{w}_0$; Loss function: $\ell$; Iterations per epoch: $T$; Epochs $E$; Freeze rate: $r^*$; Cooling time: $e^*$; Clipping bound: $C$; Momentum: $\mu$; Learning rate $\gamma$.

**for** *e = 0...E-1* **do**

    $r(e) = r^* \cdot \min(\frac{e}{e^*-1}, 1)$;

    Randomly generate a freeze mask $m \in \{0,1\}^d$ subject to $\sum m = d \cdot (1 - r(e))$;

    **for** *t = 0...T-1* **do**

        For each $x_i$ in minibatch of size $B$, compute $g_t(x_i) = \nabla \ell(w_t, x_i)$;

        Partially zero out each gradient $g_t(x_i) = m \odot g_t(x_i)$;

        Clip each individual gradient $\hat{g}_t(x_i) = g_t(x_i) \cdot \min(1, \frac{C}{\|g_t(x_i)\|_2})$;

        Add noise $\tilde{g}_t = \frac{1}{B}(\sum_i \hat{g}_t(x_i) + m \odot \mathcal{N}(0, C^2\sigma^2 \boldsymbol{I}_d))$;

        Update $v_{t+1} = \mu \cdot v_t + g_t, w_{t+1} = w_t - \gamma v_{t+1}$;

    **end**

**end**

---

active. So we present gradual cooling, which is inspired by the gradual warm-up of the learning rate adopted for non-privacy-preserving training (Goyal et al., 2017). Gradual cooling linearly ramps up the freeze rate from 0 to $r^*$ within a predefined cooling time $e^*$ and stays at $r^*$ for the remaining training epochs, i.e. $r = r^* \cdot \min(\frac{e}{e^*-1}, 1)$.

### 3.2 PER-ITERATION RANDOMIZATION VS. PER-EPOCH RANDOMIZATION

Although conducting random freeze leads to negligible additional computational cost and memory footprint, we find that it is sufficient to generate one freeze mask per epoch, while re-randomizing the freeze mask at each iteration slightly decreases the performance. As we have shown in Section 2.2, noise dominates throughout training. Therefore, by per-iteration randomization, parameters have been perturbed at this round could be frozen during subsequent iterations and stay biased. By contrast, per-epoch randomization lets the selected parameters update for one epoch, allowing the noise of multiple iterations to be averaged out.

Another strength of per-epoch randomization is that for one epoch there are certainly $(1-r)d$ parameters updated, which is favorable in collaborative learning schemes as communication cost is a significant issue, and data are not transmitted every iteration. While for per-iteration randomization, the cumulative number of updated parameters depends on the freeze rate $r$ and iterations between two communication rounds, resulting in higher communication overheads than per-epoch randomization.

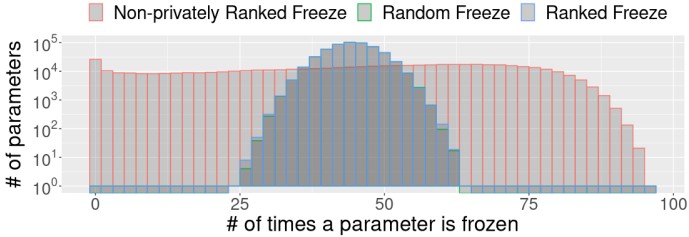

Figure 3: Histogram of the number of parameters versus the number of times a parameter is frozen. We present non-privately ranked freeze, i.e. ranking after excluding noise, for comparison.

### 3.3 RANDOM FREEZE VS. RANKED FREEZE

Based on the observation of the low-rank property of gradients (Vogels et al., 2019; Gooneratne et al., 2020; Li et al., 2020), freezing the parameters with respect to the mean of past perturbed gradients instead of a random draw might be helpful: First, taking the mean of past perturbed gradients will

average out zero-mean noise. Second, the magnitude of the gradients is a diagonal approximation to the principal components and may be indicative of a useful working subspace. We therefore consider ranking the dimensions of the gradient by their magnitude and freezing the smallest ones.

However, empirical results do not match these intuitions. Comparing Table 3 with Table 4, ranked freeze performs similarly to random freeze. We further find that ranked freeze is itself inherently random as the ranking is dominated by Gaussian noise. To demonstrate this, we run random freeze and ranked freeze[2] on End-to-end CNN for 100 epochs, then statistically analyse the distribution of how many times a parameter is frozen. The result shows the equivalency between these two strategies, which implies even averaging the perturbed gradients over a full epoch cannot sufficiently mitigate the noise added in gradient perturbation (see Figure 3). This result also reflects that even in a low privacy regime, noise has a significant impact throughout training.

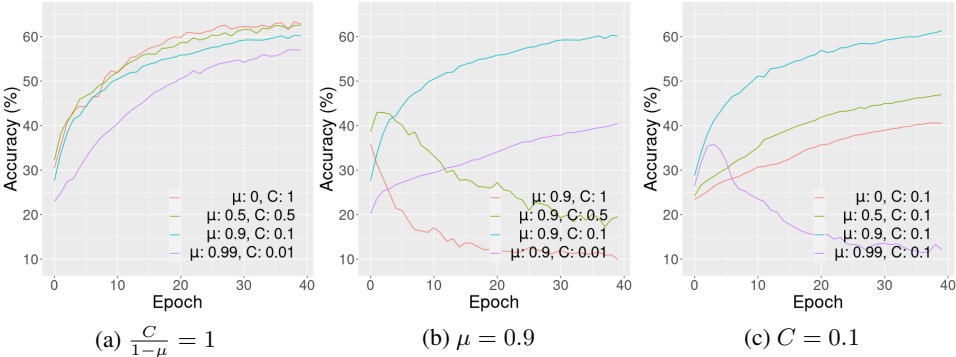

(a) $\frac{C}{1-\mu} = 1$      (b) $\mu = 0.9$      (c) $C = 0.1$

Figure 4: Test accuracy with respect to various clipping bound and momentum pairs. The network architecture is End-to-end CNN, and the privacy budget is $(\varepsilon = 3, \delta = 10^{-5})$. We adjust the clipping bound $C$ and momentum $\mu$ based on their optimal values $C = 0.1, \mu = 0.9$ (Tramer & Boneh, 2021), (a) is adjusted with respect to an inversely proportional scaling rule $\frac{C}{1-\mu} = \frac{0.1}{1-0.9} = 1$, (b) has momentum fixed $\mu = 0.9$ and varies the clipping bound, (c) has clipping bound fixed to $C = 0.1$ and varies momentum. These figures demonstrate that the inversely proportional scaling rule can achieve good performance, otherwise the network performance is degraded. We also observe that no momentum $\mu = 0$ results in better performance, probably due to less clipping bias induced by a higher clipping bound. We find out that this rule is relatively general and independent on privacy level. For experiment on low privacy budget $(\varepsilon = 7.53, \delta = 10^{-5})$, refer to Appendix F.

### 3.4 INVERSELY PROPORTIONAL SCALING RULE FOR ADJUSTING THE CLIPPING BOUND AND MOMENTUM

Previous works commonly adopt first order momentum in the optimization, because momentum can alleviate oscillation and accelerate gradient descent (Sutskever et al., 2013). As a result, it is believed to reduce the number of iterations of training and therefore achieve less privacy loss. However, for privacy-preserving training, momentum will also exaggerate the additive i.i.d. Gaussian noise by incorporating current and all historical noise. For instance, using the Pytorch (Paszke et al., 2019) implementation of SGD, the velocity update can be written as: $v_{t+1} = \mu \cdot v_t + g_{t+1}$, where $v$, $\mu$, $g$ denote perturbed velocity, momentum and perturbed gradients, respectively. Using the expression of one step noise in Equation 3 and denoting by $\hat{v}_t$ the velocity after separating the noise, we have $v_{t+1} - \hat{v}_{t+1} = (1 + \mu + \mu^2 + ... + \mu^t) \cdot \mathcal{N}(0, C^2 \cdot \sigma^2 \boldsymbol{I}_d)$. After many iterations, the scalar approximates a geometric series, i.e. $v_{t+1} - \hat{v}_{t+1} \approx \frac{1}{1-\mu} \cdot \mathcal{N}(0, C^2 \cdot \sigma^2 \boldsymbol{I}_d)$. Pulling the clipping bound $C$ out and forming the noise as $\frac{C}{1-\mu} \cdot \mathcal{N}(0, \sigma^2 \boldsymbol{I}_d)$, we present an inversely proportional scaling rule for adjusting $C$ and $1 - \mu$, i.e. with other hyperparameters fixed, networks trained with same value of the ratio $\frac{C}{1-\mu}$ perform similarly (see Figure 4). Our conjecture is that the inversely proportional scaling rule ensures the same amount of injected noise.

---

[2]For implementation details of ranked freeze, refer to Appendix C.

It is worth noticing that the inversely proportional scaling rule is general and amenable to no freeze and random freeze. Tuning hyperparameters in the context of privacy-perserving training has been observed to be brittle. This rule helps reduce the tuning workload.

## 4 EXPERIMENTAL RESULTS

**Privacy budget**  According to Definition 1, we have that each iteration of training is $(\varepsilon, \delta)$-differentially private with respect to a batch of training data, while shuffling and partitioning the dataset into batches implies that each iteration is $(O(q\varepsilon), q\delta)$-differentially private with respect to the full dataset according to the privacy amplification theorem (Balle et al., 2018; Wang et al., 2019), where $q = B/N$, $B$ is the batchsize and $N$ is the size of dataset. To track the cumulative privacy loss over multiple training epochs, we adopt Rényi differential privacy (Mironov, 2017), which is more operationally convenient and quantitatively accurate.

**Advantages of sparsity**  Projected DP-SGD induces a significant additional computation cost by running the power method and projecting gradients into and out of subspace. For the power method, the basic operation is $WW^\mathsf{T}V$, $W \in \mathbb{R}^{d \times s}$ denotes sample gradients, $V \in \mathbb{R}^{d \times b}$ denotes eigenvectors, the computational cost is $O(dbs)$. Similarly, for projection $V^\mathsf{T}X$; $X \in \mathbb{R}^{d \times n}$ denotes original gradients, the computational cost is $O(dbn)$. Applying random freeze, a random selection of rows of $X$ are deleted, while corresponding rows of $V, W$ can be removed as no information of gradient exits in that subspace. We note that $b, s$ might also be able to be reduced. Overall, the computational cost is between $O(1 - r)$ and $O((1 - r)^3)$. Another issue of projected DP-SGD is the memory footprint of $V$. Saving sparse $V$ by random freeze can be achieved by storing non-zero values and indices of zeros. The cost of indexing is logarithmic of the number of parameters, consider that $log_2 10^9 < 32$, we can decrease the memory footprint by removing a single 32 bit float gradient. Communication overhead can similarly be reduced. We note that random freeze uses the same mask during one training epoch, which could contain multiple groups of eigenvectors and communication rounds. Therefore, the cost of indexing is negligible: communication overhead and memory footprint are $\tilde{O}(1 - r)$. Further, we define the total density as the total amount of non-zero gradients by random freeze over the total amount of gradients by the original dense representation to reflect these advantages of sparsity.

Our experiments are implemented in the Pytorch framework. To compute the gradients of an individual example in a minibatch, which is required for gradient clipping, we use the BackPACK package (Dangel et al., 2020). The privacy loss of multiple iterations has been tracked with Opacus. We use the benchmark CIFAR10, which is to date standard in benchmarking DP learning. Our code is available for download from `http://anonymized.for.review/`.

To validate the reliability of random freeze, we conduct experiments on several SOTA networks cross different frameworks, including End-to-end CNN (Papernot et al., 2021); Handcrafted CNN (Tramer & Boneh, 2021) which incorporates ScatterNet (Oyallon et al., 2019) as feature extractor; Gradient Embedding Perturbation (GEP) proposed by Yu et al. (2021a) which injects noise in projected gradient; DP-Transfer Learning which adopts a pretrained network and replaces the linear classifier layer, in particular we use SIMCLR v2 (Chen et al., 2020a) pretrained on unlabeled ImageNet (Deng et al., 2009), which has been benchmarked by Tramer & Boneh (2021).

For a fair comparison, we run every experiment 5 times then compute the average of best accuracy and the standard error. We do not tune the hyperparameters when adapting the SOTA works with random freeze, instead we adopt the optimal hyperparameters for DP-SGD without random freeze provided in the respective works. Random freeze is applied as follows: if the optimal number of epochs from the previous work is set as $e^*$, we linearly ramp up the freeze rate $r$ from $r = 0$ at epoch $e = 0$ to $r = r^*$ at epoch $e = e^*$, i.e. $r = r^* \cdot \frac{e}{e^*-1}$. These straightforward experiments allow us to demonstrate that random freeze is a safe add-on in a variety of methods. We summarize the performance results in Table 1, and the corresponding total density by random freeze in Table 2. We document the hyperparameters in Appendix D.

We find that when the network is large, for instance End-to-end CNN has the most parameters among all frameworks, random freeze is able to improve the accuracy. To demonstrate this, we further tune the End-to-end CNN. We still adopt the best hyperparameters from the previous work, then first adjust the clipping bound and momentum from $(C = 0.1, \mu = 0.9)$ to $(C = 1, \mu = 0)$

| | | | Test Accuracy | | |
| --- | --- | --- | --- | --- | --- |
| Approaches | $\varepsilon$-DP | # of parameters | Baseline | Random freeze | Freeze rate $r^*$ |
| End-to-end CNN | 7.53 | 550K | $66.9 \pm 0.4$ | $66.7 \pm 0.3$ | 0.7 |
| | 3.0 | | $60.4 \pm 0.2$ | $61.1 \pm 0.2$ | 0.7 |
| Handcrafted CNN | 3.0 | 187K | $69.4 \pm 0.2$ | $69.4 \pm 0.2$ | 0.6 |
| GEP | 8.0 | 268K | $73.5 \pm 0.4$ | $73.4 \pm 0.3$ | 0.4 |
| DP-Transfer Learning | 2.0 | 41K | $92.6 \pm 0.0$ | $92.7 \pm 0.0$ | 0.7 |

Table 1: Test accuracy of SOTA works before and after adopting random freeze. We maintain the accuracy with high freeze rate. Communication overhead, computational cost and memory footprint of the projected DP-SGD are accordingly reduced as implied by total density in Table 2.

| | Total density | | |
| --- | --- | --- | --- |
| End-to-end CNN | Handcrafted CNN | GEP | DP-Transfer Learning |
| 0.65 | 0.7 | 0.8 | 0.65 |

Table 2: Total representation density of random freeze. This table is aligned with Table 1.

| | | Test Accuracy | | |
| --- | --- | --- | --- | --- |
| Approaches | $\varepsilon$-DP | Baseline | Our adjusted baseline | Random freeze |
| End-to-end CNN | 7.53 | $66.9 \pm 0.4$ | $69.7 \pm 0.1$ | $\mathbf{70.2 \pm 0.1}$ |
| | 3.0 | $60.4 \pm 0.2$ | $63.1 \pm 0.2$ | $\mathbf{64.5 \pm 0.3}$ |

Table 3: Test accuracy of End-to-end CNN adjusted with respect to inversely proportional scaling rule and trained with random freeze. Our adjusted baseline performs better than original work while with random freeze we further improve the accuracy. Altogether we obtain significantly better utility.

with respect to the inversely proportional scaling rule proposed in Section 3.4, which leads to better accuracy. Secondly, we apply random freeze. Similar to the previous experiment we also linearly ramp up the freeze rate to $r^*$ but then we extend the training for 20 additional epochs at freeze rate $r^*$, this help us to improve the performance further. The result is summarized in Table 3. Note that, after extending the training epochs, the noise multiplier $\sigma$ has been increased accordingly to ensure the same privacy budget will be consumed. As a result, more epochs without random freeze will decrease the performance. Moreover, we achieve lower total density with higher freeze rate. For hyperparameters, total density and accuracy as a function of privacy loss, refer to Appendix E.

## 5 DISCUSSION AND CONCLUSIONS

In this work we propose random freeze which is an axis-aligned random gradient dimension reduction method. We provide a fundamental theoretical study on random freeze and investigate the trade-off between signal loss and perturbation moderation. Although simple to implement, random freeze can be safely (without further tuning) applied across different frameworks and architectures. Even in conjunction with other gradient space reduction methods performance is maintained. With the sparse representation of the gradient update, random freeze can reduce the computational cost and memory footprint of projected DP-SGD. Also, it is able to reduce the total amount of transferred data in federated learning or other collaborative learning frameworks. Moreover, we significantly improve the performance of SOTA End-to-end CNN using the random freeze strategy and inversely proportional scaling rule that we have proposed. We note that the computational cost of the random freeze procedure is negligible. Therefore we believe that random freeze can be incorporated as a general method in DP-SGD. and expect random freeze to exhibit strong improvements as DP-SGD is adapted to larger and deeper networks in the future. We also hope this work can shed light on gradient dimension reduction in DP-SGD and motivate further research in this direction.

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

# A    MEAN SQUARE ERROR OF GRADIENT IN DP-SGD

**Theorem 1.** *Assuming that the gradient is drawn from $\mathcal{N}(\nabla w, \Sigma)$, centered at the true gradient $\nabla w$ and with respect to the covariance matrix $\Sigma$ whose trace goes linearly up with dimension $d$. The MSE of perturbed gradient $\tilde{g} = g + \xi_{DP}$ can be lower bounded by:*

$$MSE \geq \text{Tr}[\Sigma](1 + d\sigma^2). \tag{2}$$

*Proof.* Assuming an oracle can tell us the true gradient $\nabla w$ given a network with parameters $w \in \mathbb{R}^d$, while the individual gradient $g$ can be seen as drawn from the Gaussian distribution centered at $\nabla w$ with respect to the covariance matrix $\Sigma$, i.e. $g \sim \mathcal{N}(\nabla w, \Sigma)$. Denote $\xi_g$ as gradient deviation, i.e. $\xi_g = g - \nabla w$, the perturbed gradient $\tilde{g}$ in DP-SGD can be expressed as:

$$\tilde{g} = \nabla w + \xi_g + \xi_{DP} \tag{8}$$

and the mean square error of perturbed gradient is:

$$MSE = \mathbb{E}[\xi_g^\intercal \xi_g + 2\xi_g^\intercal \xi_{DP} + \xi_{DP}^\intercal \xi_{DP}] \tag{9}$$

Since $\xi_{DP}$ and $\xi_g$ are independent and plugging the definition of injected noise. i.e. $\xi_{DP} \sim \mathcal{N}(0, \max(g^\intercal g)\sigma^2 I_d)$, into Equation 9, we have:

$$MSE = \mathbb{E}[\xi_g^\intercal \xi_g] + \mathbb{E}[\xi_{DP}^\intercal \xi_{DP}] \tag{10}$$

$$= \text{Tr}[\Sigma] + d\sigma^2 \mathbb{E}[\max(g^\intercal g)] \tag{11}$$

$$= \text{Tr}[\Sigma] + d\sigma^2 \mathbb{E}[\max(\xi_g^\intercal \xi_g + 2\nabla w^\intercal \xi_g + \nabla w^\intercal \nabla w)] \tag{12}$$

$$\geq \text{Tr}[\Sigma] + d\sigma^2 \mathbb{E}[\max(\xi_g^\intercal \xi_g)] \tag{13}$$

where equality can be achieved when the network converges to stationary point, i.e. $\nabla w \to \mathbf{0}$.

Applying a spectral decomposition to $\Sigma$, we obtain $Q^\intercal \Lambda Q = \Sigma$, where $Q$ is an orthonormal basis and $\Lambda$ is diagonal matrix of eigenvalues of $\Sigma$. Denote by $z$ a random vector drawn i.i.d. from a normal distribution, we can further derive:

$$\xi_g^\intercal \xi_g \sim z^\intercal \Sigma z = z^\intercal Q^\intercal \Lambda Q z = z^\intercal \Lambda z \tag{14}$$

As random variable with respect to the same distribution leads to the same expectation, plugging Equation 14 into Equation 13, we have:

$$MSE \geq \text{Tr}[\Sigma] + d\sigma^2 \mathbb{E}[\max(z^\intercal \Lambda z)] \tag{15}$$

$$= \text{Tr}[\Sigma] \left(1 + d\sigma^2 \mathbb{E}\left[\max\left(\frac{1}{\text{Tr}[\Sigma]} z^\intercal \Lambda z\right)\right]\right) \tag{16}$$

$$= \text{Tr}[\Sigma] \left(1 + d\sigma^2 \mathbb{E}\left[\max\left(\frac{1}{\text{Tr}[\Lambda]} z^\intercal \Lambda z\right)\right]\right) \tag{17}$$

Since $z^\mathsf{T} \Lambda z = \lambda_1 z_1^2 + \lambda_2 z_2^2 + \cdots + \lambda_d z_p^2$; $z_i \sim \mathcal{N}(0, 1)$, we have:

$$\frac{1}{\mathrm{Tr}[\Lambda]} z^\mathsf{T} \Lambda z \sim \frac{1}{\mathrm{Tr}[\Lambda]} \sum_i Gamma\left(\frac{1}{2}, \frac{1}{2\lambda_i}\right). \tag{18}$$

So far either the exact PDF of a Gamma distribution summation can only be represented with (another) series of gamma distributions (Moschopoulos, 1985; Mathai, 1982), or its exact CDF has a series representation (Moschopoulos & Canada, 1984). For the convenience of further derivation of the expected maximum, we use the Welch-Satterthwaite approximation which provides a certain Gamma distribution:

$$\frac{1}{\mathrm{Tr}[\Lambda]} \sum_i Gamma\left(\frac{1}{2}, \frac{1}{2\lambda_i}\right) \approx \frac{1}{\mathrm{Tr}[\Lambda]} Gamma\left(\frac{\mathrm{Tr}^2\{\Lambda\}}{2\,\mathrm{Tr}\{\Lambda^2\}}, \frac{\mathrm{Tr}\{\Lambda\}}{2\,\mathrm{Tr}\{\Lambda^2\}}\right) \tag{19}$$

$$= Gamma\left(\frac{\mathrm{Tr}^2\{\Lambda\}}{2\,\mathrm{Tr}\{\Lambda^2\}}, \frac{\mathrm{Tr}^2\{\Lambda\}}{2\,\mathrm{Tr}\{\Lambda^2\}}\right) \tag{20}$$

Denote $M_n = \max(x_1, x_2, ..., x_n)$, where $x_i$ is drawn from $Gamma\left(\frac{\mathrm{Tr}^2\{\Lambda\}}{2\,\mathrm{Tr}\{\Lambda^2\}}, \frac{\mathrm{Tr}^2\{\Lambda\}}{2\,\mathrm{Tr}\{\Lambda^2\}}\right)$. While

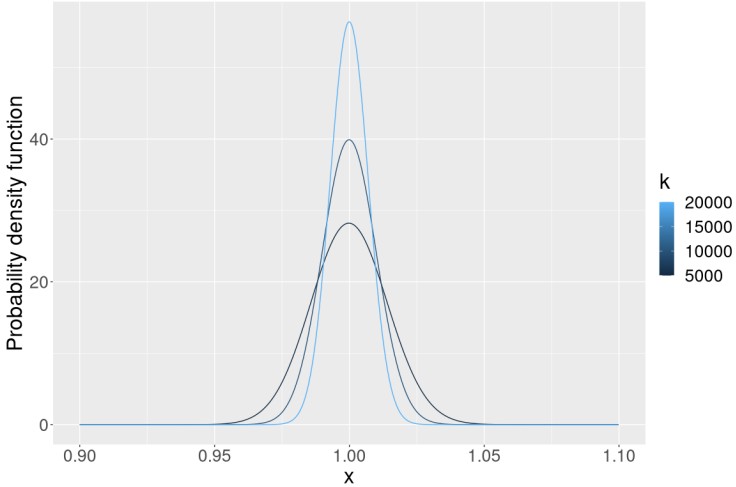

Figure 5: A series of Gamma distributions with the form $Gamma(k, k)$.

from the Cauchy-Schwarz inequality, we have:

$$|\langle \boldsymbol{I}_d, \mathrm{diag}\{\Lambda\}\rangle|^2 \leq \langle \boldsymbol{I}_d, \boldsymbol{I}_d\rangle \cdot \langle \mathrm{diag}\{\Lambda\}, \mathrm{diag}\{\Lambda\}\rangle \tag{21}$$

$$\mathrm{Tr}^2[\Lambda] \leq d \cdot \mathrm{Tr}[\Lambda^2] \tag{22}$$

$$\frac{\mathrm{Tr}^2\{\Lambda\}}{2\,\mathrm{Tr}[\Lambda^2]} \leq \frac{d}{2} \tag{23}$$

Denote $k$ as $\frac{\mathrm{Tr}^2\{\Lambda\}}{2\,\mathrm{Tr}\{\Lambda^2\}}$, Figure 5 illustrates the PDF of Gamma distribution with parameters $Gamma(k, k)$.

As $k$ becomes large, the distribution becomes more concentrated, and we have the following inequality:

$$M_n^{(1)} = \max(x_1^{(1)}, x_2^{(1)}, ..., x_n^{(1)}), x_i^{(1)} \sim Gamma(\frac{\mathrm{Tr}^2\{\Lambda\}}{2\,\mathrm{Tr}\{\Lambda^2\}}, \frac{\mathrm{Tr}^2\{\Lambda\}}{2\,\mathrm{Tr}\{\Lambda^2\}})$$

$$M_n^{(2)} = \max(x_1^{(2)}, x_2^{(2)}, ..., x_n^{(2)}), x_i^{(2)} \sim Gamma(\frac{d}{2}, \frac{d}{2})$$

$$\mathbb{E}[M_n^{(1)}] \geq \mathbb{E}[M_n^{(2)}] \geq \mathbb{E}[\frac{1}{n}\sum_i x_i^{(2)}] = 1 \tag{24}$$

Substituting inequality 24 back into Equation 17, we have:

$$MSE \geq \text{Tr}[\Sigma](1 + d\sigma^2). \tag{25}$$

The proof is completed. □

We note that $\text{Tr}[\Sigma]$ is the trace of a $d$-dimensional covariance matrix. In the extreme case that $\Sigma = \lambda I$, this will scale linearly in $d$, while in the opposite extreme of a rank deficient covariance matrix, the trace is constant in $d$. Thus, in the former case

$$MSE \geq \lambda d + \lambda \sigma^2 d^2, \tag{26}$$

while in the latter we have

$$MSE \geq \lambda + \lambda \sigma^2 d, \tag{27}$$

which leads to our conclusion that the lower bound on MSE is between linear and quadratic in $d$ in practice.

# B    THE CONVERGENCE RATE OF DP-SGD WITH RANDOM FREEZE

**Lemma 1.** *$u$, $v \in \mathbb{R}^d$ are too arbitrary vectors, $m$ is a $\{0,1\}^d$ random mask independent from $u$, $v$, and subject to $\sum m = (1-r)d$, where $r$ denotes the freeze rate. We have the following expectation:*

$$\mathbb{E}[\langle m^k \odot u, v \rangle] = (1-r)\mathbb{E}[\langle u, v \rangle]. \tag{28}$$

*Proof.*

$$\mathbb{E}[\langle m^k \odot u, v \rangle] = \mathbb{E}[\langle m \odot u, v \rangle], \tag{29}$$

$$= \sum_i \mathbb{E}[m_i]\mathbb{E}[u_i v_i], \tag{30}$$

$$= (1-r)\mathbb{E}[\langle u, v \rangle]. \tag{31}$$

The proof is completed. □

**Corollary 1.1.** *Following the notation of lemma 1, we have the expectation below:*

$$\mathbb{E}[\langle m \odot u, m \odot v \rangle] = (1-r)\mathbb{E}[\langle u, v \rangle]. \tag{32}$$

*Proof.*

$$\mathbb{E}[\langle m \odot u, m \odot v \rangle] = \mathbb{E}[\langle m^2 \odot u, v \rangle], \tag{33}$$

$$\overset{28}{=} (1-r)\mathbb{E}[\langle u, v \rangle]. \tag{34}$$

The proof is completed. □

**Lemma 2.** *$u \in \mathbb{R}^d$ is an arbitrary vector, $m$ is a $\{0,1\}^d$ random mask independent from $u$ and subject to $\sum m = (1-r)d$, where $r$ denotes the freeze rate. We have the following inequality:*

$$\mathbb{E}[\|m \odot u\|] \geq (1-r)\|u\|. \tag{35}$$

*Proof.*

$$\mathbb{E}[\|m \odot u\|] = \frac{1}{\|u\|}\mathbb{E}[\|m \odot u\|\|u\|], \tag{36}$$

$$\geq \frac{1}{\|u\|}\mathbb{E}[\|m \odot u\|^2], \tag{37}$$

$$\overset{32}{=} \frac{1}{\|u\|}(1-r)\mathbb{E}[\langle u, u \rangle], \tag{38}$$

$$= (1-r)\|u\|. \tag{39}$$

Equality is taken when $r = 0$. The proof is completed. □

**Theorem 2.** *Assume G-Lipschitz smoothness of $\nabla w$ such that $||\nabla w_{t+1} - \nabla w_t|| \leq G ||w_{t+1} - w_t||$. Consider an algorithm with clipping bound $C$, learning rate $\gamma$ and choose a symmetric probability density distribution $\tilde{p}(\cdot)$ satisfying $\tilde{p}_t(\xi_t) = \tilde{p}_t(-\xi_t), \forall \xi_t \in \mathbb{R}^d$. Then $\exists \kappa \geq 1 - r$ such that:*

$$\frac{1}{T} \sum_{t=1}^{T} P_{\xi_t \sim \tilde{p}_t}(\|\xi_t\| < \frac{C}{4}) h(\nabla w_t) \|\nabla w_t\| \leq \frac{1}{\kappa}(\frac{\Delta_{\mathcal{L}}}{\gamma T} + \gamma \Delta_C + (1-r)\gamma \Delta_{DP} - \frac{1}{T}\sum_{t=1}^{T} \mathbb{E}_m[b_t]), \quad (4)$$

*Proof.* Denote the injected noise at each iteration $\xi_{DP} := \frac{1}{B}z$, where $z$ is drawn from $\mathcal{N}(0, C^2\sigma^2 \boldsymbol{I}_d)$, follow from the smoothness assumption, we have:

$$\mathcal{L}_{t+1} \leq \mathcal{L}_t + \langle \nabla w_t, w_{t+1} - w_t \rangle + \frac{G}{2}\|w_{t+1} - w_t\|^2, \quad (40)$$

$$= \mathcal{L}_t - \gamma \langle \nabla w_t, \hat{g}_t + m \odot \xi_{DP} \rangle + \frac{G\gamma^2}{2}\|\hat{g}_t + m \odot \xi_{DP}\|^2, \quad (41)$$

$$= \mathcal{L}_t - \gamma \langle \nabla w_t, \hat{g}_t \rangle - \gamma \langle \nabla w_t, m \odot \xi_{DP} \rangle + \frac{G\gamma^2}{2}\|\hat{g}_t + m \odot \xi_{DP}\|^2, \quad (42)$$

Taking expectations on both sides and rearranging, we have:

$$\mathbb{E}[\langle \nabla w_t, \hat{g}_t \rangle] \leq \frac{1}{\gamma}\mathbb{E}[\mathcal{L}_t - \mathcal{L}_{t+1}] - \mathbb{E}[\langle \nabla w_t, m \odot \xi_{DP} \rangle] + \frac{G\gamma}{2}\mathbb{E}[\|\hat{g}_t + m \odot \xi_{DP}\|^2], \quad (43)$$

$$\stackrel{28}{=} \frac{1}{\gamma}\mathbb{E}[\mathcal{L}_t - \mathcal{L}_{t+1}] - (1-r)\mathbb{E}[\langle \nabla w_t, \xi_{DP} \rangle] + \frac{G\gamma}{2}\mathbb{E}[\|\hat{g}_t + m \odot \xi_{DP}\|^2], \quad (44)$$

$$\stackrel{28}{=} \frac{1}{\gamma}\mathbb{E}[\mathcal{L}_t - \mathcal{L}_{t+1}] - 0 + \frac{G\gamma}{2}(\mathbb{E}[\|\hat{g}_t\|^2] + \mathbb{E}[\|m \odot \xi_{DP}\|^2] - 0), \quad (45)$$

$$\stackrel{32}{=} \frac{1}{\gamma}\mathbb{E}[\mathcal{L}_t - \mathcal{L}_{t+1}] + \frac{G\gamma}{2}(\mathbb{E}[\|\hat{g}_t\|^2] + (1-r)\frac{C^2\sigma^2 d}{B^2}), \quad (46)$$

$$\leq \frac{1}{\gamma}\mathbb{E}[\mathcal{L}_t - \mathcal{L}_{t+1}] + \frac{G\gamma}{2}(C^2 + (1-r)\frac{C^2\sigma^2 d}{B^2}). \quad (47)$$

Now focusing on the l.h.s., we have:

$$\mathbb{E}[\langle \nabla w_t, \hat{g}_t \rangle] = \langle \nabla w_t, \mathbb{E}[\hat{g}_t] \rangle, \quad (48)$$

$$= \langle \nabla w_t, \frac{1}{B}\sum_i \mathbb{E}[m \odot g_{t,i} \cdot \min(1, \frac{C}{\|m \odot g_{t,i}\|})] \rangle, \quad (49)$$

$$= \mathbb{E}[\langle \nabla w_t, m \odot g_{t,i} \cdot \min(1, \frac{C}{\|m \odot g_{t,i}\|}) \rangle], \quad (50)$$

$$= \mathbb{E}[\langle m \odot \nabla w_t, m \odot g_{t,i} \cdot \min(1, \frac{C}{\|m \odot g_{t,i}\|}) \rangle], \quad (51)$$

$$= \mathbb{E}_m\left[\mathbb{E}_{\xi_t}[\langle \nabla w'_t, g'_{t,i} \cdot \min(1, \frac{C}{\|g'_{t,i}\|}) \rangle | m]\right], \quad (52)$$

$$= \mathbb{E}_m\left[\mathbb{E}_{\xi'_t \sim \tilde{p}'_t}[\langle \nabla w'_t, g'_{t,i} \cdot \min(1, \frac{C}{\|g'_{t,i}\|}) \rangle | m]\right] + \mathbb{E}_m[b_t], \quad (53)$$

where we have defined $b_t := \int \langle \nabla w'_t, g'_{t,i} \cdot \min(1, \frac{C}{\|g'_{t,i}\|}) \rangle (p'_t(\xi'_t) - \tilde{p}'_t(\xi'_t))d\xi'_t$. Equation 52 is achieved because $\xi_t$ and $m$ are independent. We note that for any given $m$, $\xi'_t$ is still independent deviation with zero mean and $\tilde{p}'_t(\xi'_t) = \tilde{p}'_t(-\xi'_t)$ since projection of symmetric distribution to subspace is symmetric. According to Theorem 2 from Chen et al. (2020b), for a given $m$ we have the following inequality:

$$\mathbb{E}_{\xi'_t \sim \tilde{p}'_t}[\langle \nabla w'_t, g'_{t,i} \cdot \min(1, \frac{C}{\|g'_{t,i}\|}) \rangle | m] \geq P_{\xi'_t \sim \tilde{p}'_t}(\|\xi'_t\| < \frac{C}{4}) \min(\|\nabla w'_t\|, \frac{3C}{4})\|\nabla w'_t\|. \quad (54)$$

Back to Inequality 47 and considering the overall $T$ steps, we have:

$$\frac{1}{T}\sum_{t=1}^{T}\mathbb{E}[\langle\nabla w,\hat{g}_t\rangle] \leq \frac{1}{\gamma T}\mathbb{E}[\mathcal{L}_1 - \mathcal{L}_T] + \frac{G\gamma}{2}(C^2 + (1-r)\frac{C^2\sigma^2 d}{B^2}), \tag{55}$$

$$\leq \frac{1}{\gamma T}\Delta_\mathcal{L} + \gamma\Delta_C + (1-r)\gamma\Delta_{DP}, \tag{56}$$

where we have defined $\Delta_\mathcal{L} := \mathbb{E}[\mathcal{L}_1 - \min_w \mathcal{L}(w)]$, $\Delta_C := \frac{GC^2}{2}$, $\Delta_{DP} := \frac{C^2\sigma^2 dG}{2B^2}$. Plugging Inequality 54 into Equation 53 and combining with Inequality 56:

$$\frac{1}{T}\sum_{t=1}^{T}\mathbb{E}_m[P_{\xi_t'\sim\tilde{p}_t'}(\|\xi_t'\| < \frac{C}{4})h(\nabla w_t')\|\nabla w_t'\|] \leq \frac{\Delta_\mathcal{L}}{\gamma T} + \gamma\Delta_C + (1-r)\gamma\Delta_{DP} - \frac{1}{T}\sum_{t=1}^{T}\mathbb{E}_m[b_t], \tag{57}$$

where we have defined $h(\nabla w_t') = \min(\|\nabla w_t'\|, \frac{3C}{4})$. Now we look at the l.h.s. and consider each step with respect to the following cases:

*case 1*: $\|\nabla w_t\| \leq \frac{3}{4}C$, then $P_m(\|\nabla w_t'\| \leq \frac{3}{4}C) = 1$,

$$\mathbb{E}_m[P_{\xi_t'\sim\tilde{p}_t'}(\|\xi_t'\| < \frac{C}{4})h(\nabla w_t')\|\nabla w_t'\|] \geq P_{\xi_t\sim\tilde{p}_t}(\|\xi_t\| < \frac{C}{4})\mathbb{E}_m[h(\nabla w_t')\|\nabla w_t'\|], \tag{58}$$

$$= P_{\xi_t\sim\tilde{p}_t}(\|\xi_t\| < \frac{C}{4})\mathbb{E}_m[\|\nabla w_t'\|^2], \tag{59}$$

$$\overset{32}{=} P_{\xi_t\sim\tilde{p}_t}(\|\xi_t\| < \frac{C}{4})(1-r)\|\nabla w_t\|^2, \tag{60}$$

$$= (1-r)P_{\xi_t\sim\tilde{p}_t}(\|\xi_t\| < \frac{C}{4})h(\nabla w_t)\|\nabla w_t\|. \tag{61}$$

*case 2*: $\|\nabla w_t\| > \frac{3}{4}C$, consider two events $A_1$: $\|\nabla w_t'\| \leq \frac{3}{4}C$; $A_2$: $\|\nabla w_t'\| > \frac{3}{4}C$ then we have:

$$\mathbb{E}_m[P_{\xi_t'\sim\tilde{p}_t'}(\|\xi_t'\| < \frac{C}{4})h(\nabla w_t')\|\nabla w_t'\|] \geq P_{\xi_t\sim\tilde{p}_t}(\|\xi_t\| < \frac{C}{4})\mathbb{E}_m[h(\nabla w_t')\|\nabla w_t'\|], \tag{62}$$

$$\overset{60}{\geq} P_{\xi_t\sim\tilde{p}_t}(\|\xi_t\| < \frac{C}{4})(P(A_1)(1-r)\|\nabla w_t\|^2 + \tag{63}$$

$$P(A_2)\frac{3}{4}C \cdot \mathbb{E}_m[\|\nabla w_t'\|]),$$

$$\overset{35}{\geq} P_{\xi_t\sim\tilde{p}_t}(\|\xi_t\| < \frac{C}{4})(P(A_1)(1-r)\|\nabla w_t\|^2 + \tag{64}$$

$$P(A_2)\frac{3}{4}C(1-r)\|\nabla w_t\|),$$

$$\geq P_{\xi_t\sim\tilde{p}_t}(\|\xi_t\| < \frac{C}{4})(1-r)\frac{3}{4}C\|\nabla w_t\|, \tag{65}$$

$$= (1-r)P_{\xi_t\sim\tilde{p}_t}(\|\xi_t\| < \frac{C}{4})h(\nabla w_t)\|\nabla w_t\|, \tag{66}$$

from which we conclude that:

$$\mathbb{E}_m[P_{\xi_t'\sim\tilde{p}_t'}(\|\xi_t'\| < \frac{C}{4})h(\nabla w_t')\|\nabla w_t'\|] \geq (1-r)P_{\xi_t\sim\tilde{p}_t}(\|\xi_t\| < \frac{C}{4})h(\nabla w_t)\|\nabla w_t\|. \tag{67}$$

The inequality above implies that $\exists\, \kappa \geq 1 - r$ such that:

$$\sum_{t=1}^{T}\mathbb{E}_m[P_{\xi_t'\sim\tilde{p}_t'}(\|\xi_t'\| < \frac{C}{4})h(\nabla w_t')\|\nabla w_t'\|] = \kappa\sum_{t=1}^{T}P_{\xi_t\sim\tilde{p}_t}(\|\xi_t\| < \frac{C}{4})h(\nabla w_t)\|\nabla w_t\|. \tag{68}$$

We note that $\kappa = 1$ if $r = 0$. Plugging equation 68 into inequality 57 we obtain:

$$\frac{1}{T}\sum_{t=1}^{T}P_{\xi_t\sim\tilde{p}_t}(\|\xi_t\| < \frac{C}{4})h(\nabla w_t)\|\nabla w_t\| \leq \frac{1}{\kappa}(\frac{\Delta_\mathcal{L}}{\gamma T} + \gamma\Delta_C + (1-r)\gamma\Delta_{DP} - \frac{1}{T}\sum_{t=1}^{T}\mathbb{E}_m[b_t]). \tag{69}$$

The proof is completed. □

It is worthwhile to note that the l.h.s. can also be written as the general form below and the inequality still holds:

$$\frac{1}{T}\sum_{t=1}^{T} P_{\xi_t \sim \tilde{p}_t}(\|\xi_t\| < zC)\min(\|\nabla w_t\|, (1-z)C)\|\nabla w_t\|, \ \forall z \in (0,1) \tag{70}$$

## C   RANKED FREEZE

Algorithm 2 describes ranked freeze, the performance is recorded in Table 4. Note that for aggregated gradient estimation we also inject noise to gradient estimation $g^e$ of frozen coordinates. If not, the coordinates get frozen at the first iteration will receive aggregated gradient estimation as 0 and never get updated for all the remaining iterations, which will degrade the network. Adding noise to frozen coordinates can give these coordinates a chance to be ranked in higher positions, while in turn the coordinates in updating but with low magnitude of true gradient may get frozen in the next iteration.

---

**Algorithm 2:** Ranked freeze

---

**Input:** Initialized parameters: $\mathbf{w}_0$; Loss function: $\ell$; Iterations per epoch: $T$; Epochs $E$; Freeze rate: $r^*$; Cooling time: $e^*$; Clipping bound: $C$; Momentum: $\mu$; Learning rate $\gamma$.

**for** *e = 0...E-1* **do**
    $g^e = \{0\}^d$;
    $r(e) = r^* \cdot \min(\frac{e}{e^*-1}, 1)$;
    **if** *e is 0* **then**
        $m = \{1\}^d$
    **else**
        Sort the indices [1,...,d] with respect to corresponding aggregated gradient of the last
        epoch $g^{e-1}$ in ascending order then set the first $d \cdot r(e)$ positions in mask $m$ to 0 and
        the rest to 1;
    **end**
    **for** *t = 0...T-1* **do**
        For each $x_i$ in minibatch of size $B$, compute $g_t(x_i) = \nabla\ell(w_t, x_i)$;
        Partially zero out each gradient $g_t(x_i) = m \odot g_t(x_i)$;
        Clip each individual gradient $\bar{g}_t(x_i) = g_t(x_i) \cdot \min(1, \frac{C}{\|g_t(x_i)\|_2})$;
        Add noise $\tilde{g}_t = \frac{1}{B}(\sum_i \bar{g}_t(x_i) + m \odot \mathcal{N}(0, C^2\sigma^2 \mathbf{I}_d))$;
        Update
        $v_{t+1} = \mu \cdot v_t + g_t, \ w_{t+1} = w_t - \gamma v_{t+1}, \ g^e = g^e + \frac{1}{B}(\sum_i \bar{g}_{t,i} + \mathcal{N}(0, C^2\sigma^2 \mathbf{I}_d))$;
    **end**
**end**

---

| Approaches | $\varepsilon$-DP | Test Accuracy | | |
| --- | --- | --- | --- | --- |
| | | Baseline | Our adjusted baseline | Random freeze |
| End-to-end CNN | 7.53 | $66.9 \pm 0.4$ | $69.7 \pm 0.1$ | $\mathbf{70.0 \pm 0.1}$ |
| | 3.0 | $60.4 \pm 0.2$ | $63.1 \pm 0.2$ | $\mathbf{64.6 \pm 0.1}$ |

Table 4: Test accuracy of End-to-end CNN adjusted with respect to inversely proportional scaling rule and trained with ranked freeze. We adopt hyperparameters recorded in Table 9. Our adjusted baseline performs better than original work while with ranked freeze we further improve the accuracy. However, we note that ranked freeze performs similarly as random freeze (see Table 3).

## D   HYPERPARAMETERS FOR TABLE 1

The $\delta$ term is $10^{-5}$ for all settings. There exist two ways of determining the injected noise scale per iteration: 1. Set the privacy budget and noise multiplier then stop the training when the privacy

budget is fully consumed; 2. Set the privacy budget and iterations of training then compute the noise multiplier so that the privacy budget will be fully consumed at the last iteration. In this work we refer to the second method when talking about hyperparameters. Hyperparameters are listed separately for each framework, see Table 5, 7, 6, 8. There is a minor difference for hyperparameters adopted from Tramer & Boneh (2021). For instance, they set batchsize=1024 for End-to-end CNN, which leads to non-exact batch partitions. We set batchsize=1000 to avoid that and achieve equally good or slightly better baseline performance. Additonally, their hyperparameters of End-to-end CNN are tuned for $\varepsilon = 3$ and then applied to $\varepsilon = 7.53$, we find out that based on their hyperparameters for batchsize, clipping bound and momentum, setting the number of epochs to 80 can achieve the best accuracy.

**End-to-end CNN**

| $\varepsilon$ | $\sigma$ | lr | Batchsize | Epoch | Momentum | Clip |
|---|---|---|---|---|---|---|
| 3 | 1.54 | 1 | 1000 | 40 | 0.9 | 0.1 |
| 7.53 | 1.10 | 1 | 1000 | 80 | 0.9 | 0.1 |

Table 5: Hyperparameters of End-to-end CNN adopted from Tramer & Boneh (2021)

**DP-Transfer Learning**

| $\varepsilon$ | $\sigma$ | lr | Batchsize | Epoch | Momentum | Clip | Source model | Aux. dataset |
|---|---|---|---|---|---|---|---|---|
| 2 | 2.30 | 4 | 1000 | 50 | 0.9 | 0.1 | SIMCLR v2 | ImageNet |

Table 6: Hyperparameters of DP-Transfer Learning adopted from Tramer & Boneh (2021)

**Handcrafted CNN**

| $\varepsilon$ | $\sigma$ | lr | Batchsize | Epoch | Momentum | Clip | Input norm | BN norm | Architecture |
|---|---|---|---|---|---|---|---|---|---|
| 3 | 5.65 | 4 | 8000 | 80 | 0.9 | 0.1 | BN | 8 | ScatterNet + CNN |

Table 7: Hyperparameters of Handcrafted CNN adopted from Tramer & Boneh (2021)

**Gradient Embedding Perturbation**

| $\varepsilon$ | lr | Batchsize | Epoch | Momentum | Clip0 | Clip1 | Subspace | Sample gradients | Aux. dataset |
|---|---|---|---|---|---|---|---|---|---|
| 8 | 0.1 | 1000 | 200 | 0.9 | 5 | 2 | 1000 | 2000 | ImageNet |

Table 8: Hyperparameters of GEP adopted from Yu et al. (2021a)

## E  IMPROVING THE ACCURACY OF END-TO-END CNN

Table 9 includes the hyperparameters and total density of ajusted End-to-end CNN with random freeze. Figure 6 shows the accuracy as a function of privacy loss.

**Adjusted End-to-end CNN with random freeze**

| $\varepsilon$ | $\sigma$ | lr | Batchsize | Epoch | Momentum | Clip | Freeze rate | Cooling time | Total density |
|---|---|---|---|---|---|---|---|---|---|
| 3 | 1.81 | 1 | 1000 | 60 | 0 | 1 | 0.9 | 40 | 0.6 |
| 7.53 | 1.18 | 1 | 1000 | 100 | 0 | 1 | 0.9 | 80 | 0.58 |

Table 9: Hyperparameters for adjusted End-to-end CNN with random freeze

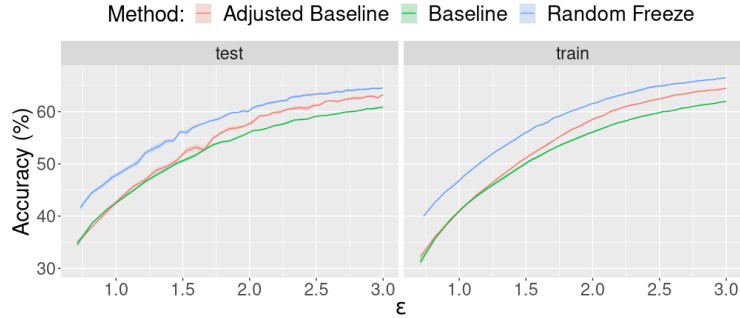

Figure 6: Accuracy as a function of privacy loss. We run five experiments and computes the mean value. The result shows that random freeze outperforms the baseline and adjusted baseline in all high privacy levels.

## F    INVERSELY PROPORTIONAL SCALING RULE

We include here an additional experiments of inversely proportional scaling rule on low privacy budget (see Figure 7).

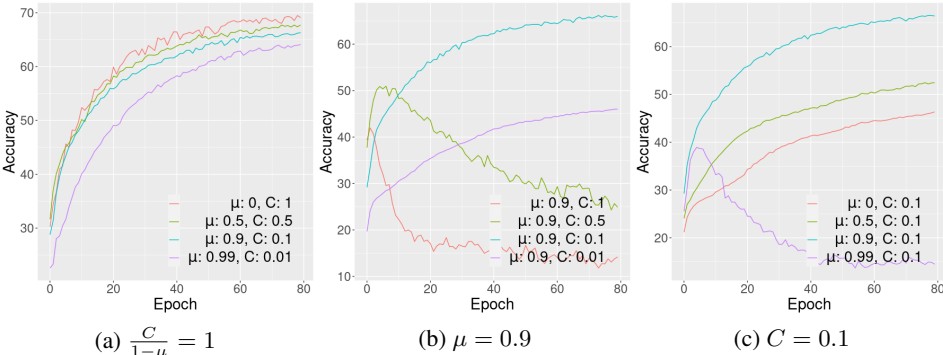

Figure 7: Test accuracy with respect to various clipping bound and momentum pairs. The network architecture is End-to-end CNN, and the privacy budget is $(\varepsilon = 7.53, \delta = 10^{-5})$. We adjust the clipping bound $C$ and momentum $\mu$ based on their optimal values $C = 0.1, \mu = 0.9$ (Tramer & Boneh, 2021), (a) is adjusted with respect to an inversely proportional scaling rule $\frac{C}{1-\mu} = \frac{0.1}{1-0.9} = 1$, (b) has momentum fixed $\mu = 0.9$ and varies the clipping bound, (c) has clipping bound fixed to $C = 0.1$ and varies momentum. These figures demonstrate that the inversely proportional scaling rule can achieve good performance, otherwise the network performance is degraded. Combining with Figure 4, we see this rule is relatively general and does not dependent on privacy level.

## G    A NUANCE OF RANDOM FREEZE

Gradient clipping is usually the first step we need to do after backpropagation in DP-SGD. However, random freeze first zeros out part of the gradient then clips with respect to the respective norm. We note that the strategy of first clipping then freezing appears in a recent work (Zhang et al., 2021) as a baseline for comparison, but shows no improvement in performance. We name this strategy posterior random freeze for convenience. For DP-SGD with posterior random freeze, denote the

gradient $\bar{\bar{g}}_t := \frac{1}{B} \sum_i m \odot g_{t,i} \cdot \min(1, \frac{C}{\|g_{t,i}\|})$, assume G-Lipschitz smoothness, we have:

$$\mathbb{E}[\langle \nabla w_t, \bar{\bar{g}}_t \rangle] \overset{43}{\leq} \frac{1}{\gamma}\mathbb{E}[\mathcal{L}_t - \mathcal{L}_{t+1}] - \mathbb{E}[\langle \nabla w_t, m \odot \xi_{DP} \rangle] + \frac{G\gamma}{2}\mathbb{E}[\|\bar{\bar{g}}_t + m \odot \xi_{DP}\|^2], \quad (71)$$

$$\overset{28}{=} \frac{1}{\gamma}\mathbb{E}[\mathcal{L}_t - \mathcal{L}_{t+1}] - (1-r)\mathbb{E}[\langle \nabla w_t, \xi_{DP} \rangle] + \frac{G\gamma}{2}\mathbb{E}[\|\bar{\bar{g}}_t + m \odot \xi_{DP}\|^2], \quad (72)$$

$$\overset{28}{=} \frac{1}{\gamma}\mathbb{E}[\mathcal{L}_t - \mathcal{L}_{t+1}] - 0 + \frac{G\gamma}{2}(\mathbb{E}[\|\bar{\bar{g}}_t\|^2] + \mathbb{E}[\|m \odot \xi_{DP}\|^2] - 0), \quad (73)$$

$$\overset{32}{=} \frac{1}{\gamma}\mathbb{E}[\mathcal{L}_t - \mathcal{L}_{t+1}] + \frac{G\gamma}{2}(\mathbb{E}[\|\bar{\bar{g}}_t\|^2] + (1-r)\frac{C^2\sigma^2 d}{B^2}) \quad (74)$$

Similarly for DP-SGP with random freeze, we have:

$$\mathbb{E}[\langle \nabla w_t, \hat{g}_t \rangle] \leq \frac{1}{\gamma}\mathbb{E}[\mathcal{L}_t - \mathcal{L}_{t+1}] + \frac{G\gamma}{2}(\mathbb{E}[\|\hat{g}_t\|^2] + (1-r)\frac{C^2\sigma^2 d}{B^2}). \quad (75)$$

We see both methods moderate the perturbation, while $\mathbb{E}[\|\bar{\bar{g}}_t\|^2] \leq \mathbb{E}[\|\hat{g}_t\|^2]$ but it is complicated to compare $\mathbb{E}[\langle \nabla w_t, \bar{\bar{g}}_t \rangle]$ and $\mathbb{E}[\langle \nabla w_t, \hat{g}_t \rangle]$, we leave this for future work. From an operational view, denote $\bar{\bar{g}}_{t,i} = m \odot g_{t,i} \cdot \min(1, \frac{C}{\|g_{t,i}\|})$ and $\hat{g}_{t,i} = m \odot g_{t,i} \cdot \min(1, \frac{C}{\|m \odot g_{t,i}\|})$, we see that $\langle \bar{\bar{g}}_{t,i}, \hat{g}_{t,i} \rangle / \|\bar{\bar{g}}_{t,i}\|\|\hat{g}_{t,i}\| = 1$ and $\|\bar{\bar{g}}_{t,i}\| \leq \|\hat{g}_{t,i}\| \leq \|g_{t,i}\|$, i.e. random freeze and posterior random freeze contain the same direction information of individual gradient, but random freeze retains more distance information. Therefore, we suppose random freeze to perform better than posterior random freeze.

