# OpenReview forum: "Differentially Private SGD with Sparse Gradients"
_ICLR.cc/2022/Conference — ICLR 2022 Submitted_

### Official Review · Reviewer_A5Uk · 2021-10-18

**Correctness:** 3
**Technical Novelty And Significance:** 2
**Empirical Novelty And Significance:** 2
**Recommendation:** 5
**Confidence:** 3

**Main Review:**

Strengths:
- A relevant problem is addressed in this work.
- The idea is simple but interesting and shows potential.

Weaknesses:
- The appendix proves a result for the MSE, but there is no insight or discussion into how this result would look for the proposed method.
- The "curse of dimentionality" is not explored in detail (i.e. performance of the method for different numbers of parameters). Relatedly, the conclusion mentions "... random freeze can reduce the computational cost and memory footprint of the power method in GEP" but this is not explored in much detail in the results.
- While the inversely proportional scaling rule is mentioned, more results would be beneficial about the choices of hyperparameters and also how robust the method is to certain parameter choices. Similarly: how many gradients can we drop before we start losing accuracy?
- In the results in Table 1 the proposed method does not improve on the accuracy and a significant accuracy improvement is also only shown for one model. It thus remains unclear if a higher DP-guarantee be obtained with higher/similar accuracy. Test accuracy as a function of privacy loss would be interesting to see.

Comments:
- A very minor comment: I am not sure I understand the positioning of section 2.5; if you see it as numerical, better to have it in the numerical section; if it is part of the discussion then let it be in the discussion.

**Summary Of The Paper:**

This paper proposes to use sparse gradients in combination with differential privacy techniques in order to mitigate performance drop that comes from applying DP to a large number of parameters.

**Summary Of The Review:**

Interesting idea but numerical experiments are not yet fully convincing and a lack of theory and discussion

---

> ### Author Response · Authors · 2021-11-15
> **Response to reviewer A5Uk**
>
> > The appendix proves a result for the MSE, but there is no insight or discussion into how this result would look for the proposed method.
>
> Theorem 1 states that DP-SGD requires a large amount of noise in its original form, and does not depend on random freezing. In our revision we will provide a fundamental theoretical study of our proposed method (Section 2). We think it should be interesting to you and alleviate your concern.
>
> > The "curse of dimentionality" is not explored in detail...
>
> The curse of dimensionality follows directly from the fact that the injected noise scales with the dimension of parameters $d$. In practice, we see that the so far largest model that has been successfully run with DP-SGD has 550K parameters (Papernot et al., 2021), which is much smaller than the majority of non-private models. Also this is in agreement with the majority of DP-SGD studies.
>
> >  the conclusion mentions "... random freeze can reduce the computational cost and memory footprint of the power method in GEP" but this is not explored in much detail in the results.
>
> The sparsity induced by random freeze is axis-aligned and the same for one training epoch. Therefore, its advantage of reducing computational cost and memory footprint can be reflected by total density, which can be theoretically proven. In our revision, we will include this analysis in the experiments section. "Total density" is included in Table 2.
>
> > While the inversely proportional scaling rule is mentioned, more results would be beneficial about the choices of hyperparameters and also how robust the method is to certain parameter choices.
>
> For hyperparameter tuning we can firstly tune either the clipping bound or momentum, then adjust these two values with respect to the scaling rule to find the best combination. Figure 2 (in revision will be Figure 4) suggests that setting momentum = 0 is a good strategy. According to our experiments the inversely scaling rule is relatively general and does not depend on the level of privacy.
>
> > how many gradients can we drop before we start losing accuracy?
>
> Table 1 shows the highest sparsity we can achieve, further sparsity will decrease the accuracy. This result is for a fixed number iterations: if we can extend training iterations (on same privacy budget), Table 2 shows we can achieve 90\% sparsity. According to our experience, gradual cooling to 90\% is a safe option among different frameworks while better accuracy has been observed on a large network, End-to-end CNN.
>
> > In the results in Table 1 the proposed method does not improve on the accuracy and a significant accuracy improvement is also only shown for one model.
>
> The goal of Table 1 is to show the highest sparsity we can achieve with random freeze while maintaining the accuracy, while training with the same number of iterations. Table 1 shows that applying random freeze does not decrease performance, while providing other advantages of sparsity.
>
> >  It thus remains unclear if a higher DP-guarantee be obtained with higher/similar accuracy. Test accuracy as a function of privacy loss would be interesting to see.
>
> 1) We note that in some previous work there is a privacy loss and accuracy plot, however it is usually a plot of iterations and accuracy while the iteration axis is represented by the accumulated privacy loss. So hyperparameters are tuned for the total number of iterations.  According to our experiments, the optimal hperparameters, especially $\sigma$, are not the same for different privacy budgets. Tuning hyperparameters for every level of privacy is a large overhead: instead previous works report the utility at certain privacy levels. We adopt the reported optimal hyperparameters in previous works and compare them with random freeze, so we could not plot the real relationship between privacy loss and accuracy as the complete optimal hyperparameter set is not provided by the previous works.
>
> 2) Overall we think our comparison is convincing. For the concern about how random freeze performs for high privacy, we will theoretically show in our revision that random freeze moderates the perturbation. For high levels of privacy, we usually need to inject more noise, i.e. higher perturbation, so random freeze is expected to perform better. Table 2 also reflects that random freeze gains more accuracy for higher $\epsilon$.

---

> > ### Author Response · Authors · 2021-11-17
> > **Response to reviewer A5Uk**
> >
> > For the concern about accuracy as a function of privacy loss:
> >
> > We upload the result of accuracy as a function of privacy loss (Figure 6). Similarly to previous work, we set here $\sigma$ to a constant for all $\epsilon$. As we can see, random freeze also outperforms the baseline in high privacy levels.
> >
> > For the concern about inversely proportional scaling rule:
> >
> > In our revision, we show the result of the inversely proportional scaling rule for a low privacy level (Figure 7). According to our experiments, this rule is relatively general. We hope the additional result has alleviated your concern.

---

### Official Review · Reviewer_omCe · 2021-11-01

**Correctness:** 3
**Technical Novelty And Significance:** 3
**Empirical Novelty And Significance:** 4
**Recommendation:** 5
**Confidence:** 3

**Main Review:**

Strengths:
The method boasts empirical success
Paper is well-written

Weakness:
The proposed method, although reasonable, seems rather ad hoc. Thus, without a guiding first-principles motivation for the method, one should wonder how generally the method works and why it works.
The paper does not do a sufficiently thorough job at the systems considerations of the work.


**Summary Of The Paper:**

This work introduces the idea of randomly pruning gradients in order to reduce the dimensionality, which, in turn, results in greater efficiency for DP-SGD. The work provides an empirical analysis to support the method.

**Summary Of The Review:**

Overall, this method seems purely heuristic. Without a theoretical or first-principles explanation, this paper relies mostly on an empirical analysis. Theorem 1 is included, but the Gaussian assumption for SGD is extremely limiting and deviates from the accepted standards of analysis for SGD, which generally assume unbiased estimates with bounded variance, but not Gaussian shape in particular. Furthermore, the connection between Theorem 1 and the proposed method of random freezing does not seem that solid.

For the systems analysis, the authors do not report two key quantities: (1) the memory consumption (in bytes) of the method and (2) the runtime of the method. For (1), sparse representations must generally pay a memory overhead because an index is also required. A sparsity % of X% does not imply X% of the memory used. For (2), it seems that although a sparse representation can reduce memory, I am not sure how the runtime of the sparse gradient computations might also be affected here in the DP-SGD. It is important to also benchmark and report this, or at least mention that the runtime is not significantly affected if that is the case.

To summarize, the work is neither a complete systems analysis nor provides a sufficient fundamental analysis. I feel that at least one of these two aspects should be present.

---

> ### Author Response · Authors · 2021-11-15
> **Response to reviewer omCe**
>
> >Overall, this method seems purely heuristic. Without a theoretical or first-principles explanation, this paper relies mostly on an empirical analysis. Theorem 1 is included, but the Gaussian assumption for SGD is extremely limiting and deviates from the accepted standards of analysis for SGD, which generally assume unbiased estimates with bounded variance, but not Gaussian shape in particular. Furthermore, the connection between Theorem 1 and the proposed method of random freezing does not seem that solid.
>
> We assume a Gaussian distribution of the gradient in order to estimate the maximum norm of the gradient. Theorem 1 is a prior theorem saying DP-SGD requires a very large amount of noise in its original form, it is not connected with random freeze. However, in our revision we will provide a fundamental theoretical study on random freeze, we find that random freeze exhibits a trade-off between signal loss and perturbation moderation in DP-SGD. We believe this is a fundamental result of general interest.
>
> >  (1), sparse representations must generally pay a memory overhead because an index is also required. A sparsity % of X% does not imply X% of the memory used. (2), it seems that although a sparse representation can reduce memory, I am not sure how the runtime of the sparse gradient computations might also be affected here in the DP-SGD...
>
> We note that sparsity induced by random freeze is axis-aligned and is the same for one training epoch, which makes the cost of saving or transferring indices negligible. So the memory footprint and communication overhead is approximately proportional to density $1-r$. We discuss the computational cost of the power method and projection in low-rank DP-SGD, not forward and backward propagation, if that is what the reviewer thought. In our revision, we will provide a theoretical proof of these advantages in the experiments section.

---

> > ### Comment · Reviewer_omCe · 2021-11-28
> > **reviewer response**
> >
> > Thank you for the rebuttal.
> >
> > I appreciate the clarification about axis-alignment for the sparsity. That indeed removes the need for a separate (i.e. CSR) indexing.
> >
> > As for the new theoretical results, while I appreciate the high-level ideas, I find Thm. 2 to be somewhat opaque. I feel that the analysis could be refined and distilled into something nice, but that it requires a bit more massaging.
> >
> > Overall, I am trending towards improving my score to a 6, but I still feel that the theoretical analysis could benefit from further work, so I might keep my score.

---

> > > ### Author Response · Authors · 2021-11-29
> > > **Response to reviewer omCe**
> > >
> > > Thank you for your reply. We appreciate that the paper can benefit from a clearer statement significance of Theorem 2. It actually clearly reveals why the random freeze strategy can maintain or even further improve the performance of DP-SGD. First, for vanilla SGD without injected noise, random freeze will impede the convergence, it is able to be theoretically proven that the gradient norm decreases slower by assuming Lipschitz-smoothness. This implies that if we limit the number of training iterations as required by DP-SGD, then we will end up with a worse network by applying the random freeze strategy to vanilla SGD without noise.
> > >
> > > In contrast, also under the assumption of Lipschitz-smoothness, Theorem 2 implies that by applying random freeze to DP-SGD, the speed of convergence does not necessarily become slower, it depends on a trade-off between the term induced by injected noise and the others as the noise term is reduced and other terms are increased. Since random freeze removes the noise and the gradient information of a subset of coordinates, from this operational view we can readily understand that this is a trade-off between the signal loss and perturbation moderation. In the section following Theorem 2, we have empirically demonstrated that the injected noise dominates the gradient information on a relatively large network. So combining this with Theorem 2, we show that it is possible to maintain or accelerate convergence by applying random freeze to DP-SGD.
> > >
> > > To conclude, Theorem 2 completes the theoretical analysis of the random freeze strategy. We understand the reviewer's concern, we hope our additional explanation has alleviated it. We will update the final version of our paper with a more clear explanation of the significance.

---

### Official Review · Reviewer_w3pG · 2021-11-01

**Correctness:** 2
**Technical Novelty And Significance:** 2
**Empirical Novelty And Significance:** 2
**Recommendation:** 3
**Confidence:** 4

**Main Review:**

This paper proposes a dimension reduction technique for private machine learning called random freeze. The basic idea is to randomly select a subset of model parameters and zero out corresponding gradients during the training process. Another contribution is a method to counter the residual effect of added noise when momentum is used for gradient updates.

Strength:
+ The problem the paper aims to address is an important one. Machine learning with differential privacy is a very active research area, and this paper is a valuable addition to the literature in this field.

+ The proposed method is simple to understand and implement. It is also general and applicable to all types of neural networks. If proven effective, it could be adopted into many existing frameworks to improve the utility private machine learning.

Weakness:
- The experiments do not sufficiently demonstrate the benefits of the proposed method. As a dimension reduction technique, random freeze could be compared with other similar techniques (e.g., [1]) for private learning in terms of utility. Among the four models used in the experiments, only one (GEP) is a belongs to this category, and random freeze shows a drop in utility (from 73.5 to 73.4, third row of table 1). As such, it is unclear if random freeze can be used in place of, or together with other dimension reduction techniques.

- The proposed scaling of gradient clipping bound is not convincing. The paper argues that, because of the use of momentum in gradient updates, the noise added in previous iterations has residual effects in later iterations. Therefore, the noise scale needs to be adjusted to counter this effect. In the paper, the way to adjust the noise is to use a single constant to scale all noises, and this constant is inverse to the cumulative effect when the number of iterations go to infinity. However, this method for adjusting the noise does not make sense to me. For example, the noise added in the first iteration of training would have much more cumulative effect than the noised added in the final interaction. Also, due to random freeze, fewer and fewer (and different) parameters are receiving updates as the training progress. Thus, while the concern raised by the paper is a valid one, I am not sure the proposed method is the right solution.

- The paper claims (in Section 1.2, Our Contribution) that the proposed method “reduce the computational cost and memory footprint induced by the power method”. However, this claim is neither theoretically analyzed nor empirically demonstrated using experiments.

- Discussion on related work is missing.


Typos:
Page 8 footnote: “There exist too ways…”-> “There exist two ways…”
Page 9 last paragraph: “It is interested to observe…” -> “It is interesting to observe…”

References:
[1] LoRA: Low-Rank Adaptations of Large Language Models.


**Summary Of The Paper:**

The paper aims to improve the effectiveness of deep learning under differential privacy. It proposes a simple dimension reduction method that could potentially improve model accuracy as well as mitigate the computational and memory overhead of some previous power method.

**Summary Of The Review:**

This paper has two core contributions: a dimension reduction technique (random freeze) and a method to scale gradient clipping norm. However, as mentioned in the discussion on weakness, this paper does not sufficiently demonstrate the benefits of the proposed methods. Therefore, it does not meet the standard for publication at a top conference like ICLR in my opinion.

---

> ### Author Response · Authors · 2021-11-15
> **Response to reviewer w3pG**
>
>   > As a dimension reduction technique, random freeze could be compared with other similar techniques (e.g., [1]) for private learning in terms of utility.
>
> 1) We note that LoRA has also been submitted to ICLR 2022 and it appeared on preprint platform within last four months. According to ICLR policy (https://iclr.cc/Conferences/2022/ReviewerGuide), this is considered "very recent work" and should be disregarded in the evaluation of this submission. Nevertheless, we also note that LoRA is an approach for non-private adaptation of pre-trained language models. The utility of LoRA with DP-SGD is not reported in its paper. An independent study on its utility in differentially private training is out of the scope of our work.
>
> 2) For low-rank optimization methods, differential privacy prevents our access to unperturbed private gradients or the model, and limits training iterations. To the best of our knowledge, low-rank optimization under privacy guarantees has only recently been studied, e.g. in GEP.  GEP has a code release and is SOTA on the CIFAR10 benchmark. Therefore, we believe our experiment with GEP is sufficient in this subfield.
>
> 3) Random freeze reduces the gradient dimension but it does not extract any characteristic information from gradient or model it also does not necessarily rely on a low-rank assumption. So random freeze is orthogonal to low-rank DP-SGD and previous works. We have provided a theoretical study on this in our revision.
>
> > random freeze shows a drop in utility (from 73.5 to 73.4, third row of table 1)...
>
> We do not agree the reviewer saying that using random freeze with GEP leads to utility drop as avg. test accuracy goes from 73.5\% to 73.4\%. This difference is statistically insignificant. In contrast, we significantly improve the performance of large network, i.e. SOTA End-to-end CNN, using the random freeze strategy and the inversely proportional scaling rule that we have proposed by 4.1\% and 3.3\% for $\epsilon = 3$ and $\epsilon = 7.53$, respectively. Furthermore, in consideration of other benefits brought by sparsity, the advantages of random freeze are clearly shown.
>
> >... this method for adjusting the noise does not make sense to me. For example, the noise added in the first iteration of training would have much more cumulative effect than the noised added in the final interaction. Also, due to random freeze, fewer and fewer (and different) parameters...
>
> We want to point out that it is our intention to make the scaling rule of the clipping bound and momentum have a simple form. Indeed the cumulative effect is different for the first iteration and the final iteration, but the total error of approximation is not large. Define the approximation error $e$ as the difference between actual cumulative effect by running $k$ steps and take the limit of a geometric series.  We have $e = \mu^k$, where $\mu$ is the momentum factor. Note that this error decays exponentially. For the experiment we have conducted, there are 2000 iterations in total (which is relatively small). Assuming $\mu = 0.99$, we have $\mu^{300}\approx0.049$, so 85\% of iterations have approximation error less than 5\%. While if we assume $\mu=0.9$, which is more realistic in practice, we have that more than 98\% of iterations achieve an approximation error less than 5\%. For random freeze, we zero out gradient rather than momentum so the cumulative effect should be the same. Importantly, our experiments, demonstrated in Figure 2 (Figure 4 in revision), clearly show the effectiveness of our scaling rule.
>
> > ...the proposed method  “reduce the computational cost and memory footprint induced by the power method”. However, this claim is neither theoretically analyzed nor empirically demonstrated using experiments.
>
> The sparsity induce by random freeze is axis-aligned, so its advantage of memory footprint and computational cost can be straightforwardly theoretically proven. We have incorporated this proof in our experiments section.
>
> > Discussion on related work is missing.
>
> Our discussion of related work was incorporated in the introduction section 1.1 for the flow of the article.  We have also improved the related work section in our revision.
>
> > Typos: Page 8 footnote: “There exist too ways…”-> “There exist two ways…” Page 9 last paragraph: “It is interested to observe…” -> “It is interesting to observe…”
>
> We have fixed the typos, thank you for that.

---

> > ### Author Response · Authors · 2021-11-17
> > **Response to reviewer w3pG**
> >
> > For the concern about inversely proportional scaling rule:
> >
> > We have uploaded another experiment testing this rule in low privacy budget (Figure 7). According to our experiments, this rule is robust and effective. We hope the additional experiment has alleviated your concern.

---

### Official Review · Reviewer_rw2E · 2021-11-09

**Correctness:** 3
**Technical Novelty And Significance:** 2
**Empirical Novelty And Significance:** 3
**Recommendation:** 5
**Confidence:** 4

**Main Review:**

I think this paper has made the following contributions:
1. This paper proposes a new algorithm of differentially privately training neural networks, which randomly freezes a progressively increasing subset of parameters. This light-weight algorithm can be easily implemented, which dramatically reduces the communication cost while maintaining the accuracy. The idea of gradual cooling and inversely proportional scaling rule are also very interesting.
2. The paper justifies its statement with detailed experiments.

I think the paper can be improved in the following ways:
1. First, the idea of the random freezing is not complex. Therefore, I am not sure how much algorithmic contribution it has made. Besides, this idea is not fully novel, at least it has appeared in other papers. For example, in this paper (https://arxiv.org/pdf/2103.01294.pdf), I remember they also used some similar approach in the experiment section.
2. Personally I am very curious about the performance comparison between random freeze and ranked freeze, which is not provided in this paper. The paper justifies why they do not adopt the ranked freeze in section 2.3, which I do not fully follow. I do not think it is enough to just show they have a similar histogram. Consider the following example, in each iteration, each coordinate is 1 with prob 0.5; else 0. Note that the random freeze and ranked freeze with constant gamma = 0.5 give you exactly the same histogram. However, there is much more information left in the ranked freeze.
3. Some related work is missing, at least the paper I just mentioned, and this paper (https://scholar.google.com/citations?view_op=view_citation&hl=en&user=m8NUgw0AAAAJ&sortby=pubdate&citation_for_view=m8NUgw0AAAAJ:u_35RYKgDlwC).



**Summary Of The Paper:**

This paper considers the problem of differentially privately learning deep neural networks. In order to improve the accuracy and reduce the communication cost, this paper proposes to randomly freeze a progressively increasing subset of parameters, which results in sparse gradient updates. Empirical results also show that the new algorithm can largely reduce the communication cost, while maintaining the performance. Furthermore, the extra computation cost is negligible.

**Summary Of The Review:**

Overall, I think it is a borderline paper. So I am fine with either accepting or rejecting this paper.

---

> ### Author Response · Authors · 2021-11-15
> **Response to reviewer rw2E**
>
> >...this idea is not fully novel, at least it has appeared in other papers. For example, in this paper (https://arxiv.org/pdf/2103.01294.pdf), I remember they also used some similar approach in the experiment section.
>
> 1. The work by Zhang et. al. proposes to select top-k gradients via DP selection, they use a random strategy as a trivial baseline while showing that the baseline could not outperform DP-SGD. We note that in their algorithm they conduct a clipping first then apply freezing, while random freeze firstly zeros out the gradient then clips. This nuance could lead to a difference in behaviour, and we have discussed this in Appendix G in our revision. We will upload the revision soon.
>
> 2. In our revision, we provide a fundamental theoretical study (Section 2) which shows that random freeze leverages the trade-off between signal loss and perturbation moderation in DP-SGD. So random freeze does not necessarily rely on a low-rank assumption and is orthogonal to previous works. To the best of our knowledge, we are the first to study this issue and propose an effective approach.
>
> >...Consider the following example, in each iteration, each coordinate is 1 with prob 0.5; else 0. Note that the random freeze and ranked freeze with constant gamma = 0.5 give you exactly the same histogram. However, there is much more information left in the ranked freeze.
>
> 1) First, we would like to show that in the reviewer's example, ranked freeze and random freeze will have similar histograms, but ranked freeze does not contain more information than random freeze: inherently they are still the same. To compute any statistics from the perturbed gradient, it is common to use accumulated historical gradients, otherwise for a single perturbed gradient the ranking is (nearly) random due to the large amount of noise. However, in the reviewer's example, the expectation of accumulated historical gradients is 0.5 for every coordinate, so ranking and selecting the top-k is the same as randomly selecting k elements, i.e. ranked freeze performs inherently the same as random freeze. Second, the example assumes the same mean and variance for each coordinates, which does not match the characteristic of networks. In networks, there should be a group of coordinates having a larger mean gradient. Although the mean value of the gradient of each coordinate evolves during training, if we can rank with respect to the unperturbed gradients then in practice we should see that a group of coordinates always or usually stays at the top of the ranking list, and some of the coordinates usually get frozen. To demonstrate this (in our revision), we run non-privately ranked freeze with respect to unperturbed gradients and draw the histogram. The Figure shows clearly that ranked freeze performs similarly to random freeze rather than non-privately ranked freeze, which indicates privately ranked freeze is inherently random.
> 2) Actually, we ran experiments and saw that both strategies had similar performance at first. After studying the results, we realized that ranked freeze performs inherently randomly. So we use the histogram to reflects this insight in our work. We agree that just giving the histogram does not convey our full insights. We have revised Section 2.3 (Section 3.3 in revision) and attached the results of performance in our revision. You can compare Tables 3 and 4 in our revision.
>
> >Some related work is missing, at least the paper I just mentioned, and this paper (https://scholar.google.com/citations?view_op=view_citation&hl=en&user=m8NUgw0AAAAJ&sortby=pubdate&citation_for_view=m8NUgw0AAAAJ:u_35RYKgDlwC).
>
> Thank you for providing us two interesting related works. We have incorporated them in our revision and also improved the related works section.

---

> > ### Comment · Reviewer_rw2E · 2021-11-23
> > **Response**
> >
> > I have read the rebuttal. However, I am still not fully persuaded by the authors. So I decide to keep my score unchanged.

---

### Author Response · Authors · 2021-11-17
**Rebuttal revision**

We thank the reviewers for their helpful comments and insightful reviews. With your feedback, we are able to provide a better work. In this revision we have added:

1) A theoretical study of random freeze (Section 2). Theorem 2 points out that random freeze essentially leverages the trade-off between signal loss and perturbation moderation in DP-SGD. We remark that our theoretical analysis does not necessarily rely on a low-rank assumption. From an operational view, random freeze does not extract any characteristic information of gradient or model, or approximate them from a subspace. Therefore, our work is orthogonal to previous works. To the best of our knowledge, we are the first to study this issue and propose an effective approach.
2) A figure of accuracy as a function of privacy loss (Figure 6). We show that random freeze outperforms the baseline at a high privacy level.
3) A figure of the inversely proportional scaling rule on a low privacy budget (Figure 7), we show that this rule is relatively robust and effective at different privacy levels.
4) A section discussing the difference between first clipping then random freeze and first random freeze then clipping (Appendix G).
5) A paragraph in the experiments section to theoretically analyze the advantages of sparsity.

We have also improved:
1) The related works section.
2) The figure for random freeze vs. ranked freeze.

---

> ### Author Response · Authors · 2021-11-23
> **Rebuttal revision**
>
> We have fixed some typos in our second revision.

---

### Decision · Program_Chairs · 2022-01-20

**Decision:**

Reject

**Comment:**

This paper provides a new differentially private training method. The key idea is sparse gradient updates---that is, their variant of differentially private SGD (DP-SGD) only updates on a random subset of the parameters in each iteration. The authors argued that their method has a benefit in terms of memory and communication efficiency. The reviews suggested that the paper may require further evidence to motivate and justify the novelty of the proposed method. First, the reviewers are not fully convinced that the proposed method reduced both memory and communication. In particular, would the technique of random freeze require running DP-SGD for more iterations? Even though the authors added a new theoretical result (mostly adapted from Chen et al.), the newly added Theorem 2 does not explain the benefits of the freezing technique. Thus, the paper can benefit from more extensive theoretical analyses or justification. The authors should also consider including the additional related work brought up by the reviewers. In summary, the paper is not ready for publication at ICLR.